# Neurosonological Findings Related to Non-Motor Features of Parkinson’s Disease: A Systematic Review

**DOI:** 10.3390/brainsci11060776

**Published:** 2021-06-11

**Authors:** Cristina del Toro Pérez, Laura Amaya Pascasio, Antonio Arjona Padillo, Jesús Olivares Romero, María Victoria Mejías Olmedo, Javier Fernández Pérez, Manuel Payán Ortiz, Patricia Martínez-Sánchez

**Affiliations:** 1Neurosonology Laboratory, Department of Neurology, Torrecárdenas University Hospital, 04009 Almería, Spain; cristinadeltoro@msn.com (C.d.T.P.); laura.amaya.pascasio@gmail.com (L.A.P.); aarjonap@gmail.com (A.A.P.); olivares.je@gmail.com (J.O.R.); v.mejiasolmedo@hotmail.com (M.V.M.O.); javifp1985@gmail.com (J.F.P.); payanortiz@hotmail.com (M.P.O.); 2Faculty of Health Sciences, CEINSA (Center of Health Research), University of Almería, La Cañada, 04120 Almería, Spain

**Keywords:** transcranial sonography, Parkinson’s disease, non-motor symptoms, systematic review, depression, apathy, autonomic dysfunction, bladder dysfunction, restless legs syndrome, sleep disorders, cognitive disorders, dementia, hallucinations, apathy

## Abstract

Non-motor symptoms (NMS) in Parkinson’s disease (PD), including neuropsychiatric or dysautonomic complaints, fatigue, or pain, are frequent and have a high impact on the patient’s quality of life. They are often poorly recognized and inadequately treated. In the recent years, the growing awareness of NMS has favored the development of techniques that complement the clinician’s diagnosis. This review provides an overview of the most important ultrasonographic findings related to the presence of various NMS. Literature research was conducted in PubMed, Scopus, and Web of Science from inception until January 2021, retrieving 23 prospective observational studies evaluating transcranial and cervical ultrasound in depression, dementia, dysautonomic symptoms, psychosis, and restless leg syndrome. Overall, the eligible articles showed good or fair quality according to the QUADAS-2 assessment. Brainstem raphe hypoechogenicity was related to the presence of depression in PD and also in depressed patients without PD, as well as to overactive bladder. Substantia nigra hyperechogenicity was frequent in patients with visual hallucinations, and larger intracranial ventricles correlated with dementia. Evaluation of the vagus nerve showed contradictory findings. The results of this systematic review demonstrated that transcranial ultrasound can be a useful complementary tool in the evaluation of NMS in PD.

## 1. Introduction

Parkinson’s disease (PD) is a chronic progressive neurodegenerative disorder characterized not only by its motor aspects, but also by numerous non-motor symptoms (NMS) that encompass neuropsychiatric manifestation, sensory abnormalities, behavioral changes, sleep disturbances, and autonomic dysfunction. NMS may be the presenting clinical feature of PD in over 20% of individuals, which usually delays PD diagnosis and an early appropriate treatment [1]. Various studies have demonstrated that NMS have a greater impact on quality of life than motor manifestations, even during the first years after diagnosis. Moreover, hallucinations have been pointed out as the strongest predictor of nursing home placement for people with PD [2].

Depression and apathy are common in PD, with 40% of patients presenting apathy and 17% suffering from a major depressive disorder, occurring at any time during the course of the disease [3,4]. Common autonomic complaints are orthostatic hypotension, gastrointestinal dysfunction, and urinary symptoms. Together with REM sleep behavior disorder (RBD), they present a prevalence in the range of 25–50% [1].

During the last decades, there has been a growing use of transcranial sonography (TCS) to evaluate brainstem and subcortical brain structures as a complementary tool in the diagnosis of PD. TCS is reliable and sensitive in detecting basal ganglia abnormalities and has proven its potential to identify idiopathic PD from healthy controls based on substantia nigra (SN) hyperechogenicity, which is present in 67 to 95% PD patients compared to 3 to 9% in subjects without PD [5,6,7].

A review of TCS findings associated to NMS in PD performed by Walter et al. in 2010 showed evidence that some midbrain changes may be related to NMS and can contribute to their identification [8]. Since then, several studies exploring this topic have been published and new techniques have been developed.

The aim of the systematic review is to provide a clear view on the most relevant abnormalities identified with TCS and other ultrasound techniques that can be related to the presence of NMS in PD. The main NMS addressed are depression, anxiety, apathy, hallucinations, cognitive disorders, autonomic dysfunction, restless legs syndrome, sleep disorders, pain, fatigue, anosmia, ageusia, and libido alterations.

## 2. Materials and Methods

### 2.1. Search Strategy

This protocol follows the guidelines according to the preferred reporting items for systematic reviews and meta-analysis protocol (PRISMA-P) [9]. It was registered in the PROS PERO international database of prospectively registered systematic reviews (CRD 42021250195). PubMed, Scopus, and Web of Science electronic databases were searched for articles in English or Spanish, published up to January 2021, and with the following criteria: cross-sectional, case-control, and cohort observational studies including patients with Parkinson’s disease, ultrasound assessment of neurological structures and evaluation of NMS, analyzing differences between echogenicity and/or size of the evaluated structures between PD suffering or not from a specific NMS. Case reports were excluded. The search query was: (“non-motor symptoms” OR “depression” OR “fatigue” OR “low blood pressure” OR “autonomic dysfunction” OR “orthostatic hypotension” OR “bladder dysfunction” OR “restless legs syndrome” OR “sleep disorders” OR “REM-sleep behavior disorder” OR “pain” OR “cognitive disorders” OR “anxiety” OR “hallucinations” OR “delusions” OR “anosmia” OR “apathy” OR “ageusia” OR “libido” OR “constipation”) AND (“Parkinson’s disease” OR “PD”) AND (“transcranial sonography” OR “ultrasound” OR “transcranial ultrasonography”).

In addition to the database search, a manual revision of the reference lists of all relevant articles was performed to identify additional studies of interest.

### 2.2. Selection of Studies

Two researchers (C.T. and L.A.) separately reviewed the titles and abstracts of the retrieved articles to determine the presence of the abovementioned criteria. Disagreements were solved by the consensus of a third author (P.M.). Duplicated entries, studies on diseases different from PD or evaluation techniques other than ultrasound, papers not written in English or Spanish, publications that were not research studies, and any other article that did not fit with the scope of the review were excluded.

### 2.3. Data Extraction

Upon manuscript selection, the following information was extracted: the number of participants and socio-demographic characteristics, the assessed NMS and the evaluation protocol or diagnostic strategies, the ultrasound modalities, and the major findings reported.

A limited number of studies were expected to be found by the systematic search and they were expected to be clinically and methodologically heterogeneous. Likewise, some of the results were based on qualitative findings. Therefore, conducting a meta-analysis was not included in this protocol.

### 2.4. Quality Assessment

The risk of bias of the included studies was evaluated using QUADAS-2 [10] for assessing the risk of bias recommendations by The Cochrane Collaboration. In this review, there is no gold standard test for comparison of the ultrasound findings. Consequently, we considered the proposed diagnosis criteria of the non-motor syndrome, based on validated scales or neurologist advice, for each study as the reference gold standard.

## 3. Results

After removing duplicates, the database search yielded 263 results. An additional 14 studies were identified through the references of the principal records. A total of 277 publications were screened for eligibility and 254 studies were excluded for the following reasons: publications involving different pathologies, symptoms evaluated in a population different from PD, systematic reviews, and animal experimental studies. The PRISMA Flow Diagram is shown in Figure 1. Eventually, 23 studies were included and are summarized in Table 1.

### 3.1. Study Characteristics

The included studies were published between 1997 and 2020, with 60% published during the last five years. The articles consisted of cross-sectional, case-control, and cohort prospective studies, including mainly patients with PD, healthy controls, and non-PD patients with depression or other NMS. The mean participant sample size was 133 (SD = 85.2; range = 81–143). The PD participants’ ages ranged from 45 to 77, with a majority being male patients. The disease duration varied considerably between studies and within each study, ranging from 30 months to 15 years. Three studies included newly diagnosed PD. The main NMS evaluated was depression (12 articles), followed by dementia (4 studies), and dysautonomic symptoms (4 studies). Standardized clinical scales and neurologist or psychiatrist evaluation were the preferred instrument used for assessing NMS, summarized in Table 1. Main referred structures identified by transcranial ultrasound can be found in Appendix A.

### 3.2. Quality Assessment

We analyzed the quality of the studies using the QUADAS-2 tool. Most of the observational studies showed a low risk of bias. Regarding patient selection, the main limitations were that the sample was not based in an epidemiological registry and in a few studies, selection criteria were not clearly described. For the index test, most studies were homogeneous, describing the pre-stablished evaluation criteria, with more than one experienced evaluator blinded to the patient diagnosis. In addition, inconsistent application of reference standard and not having a clear time of application were identified (Figure 2).

### 3.3. Main Findings

#### 3.3.1. Depression

In 1997, Becker el al. [11] evaluated for the first time ultrasound midbrain changes in depressed PD patients, comparing with non-depressed PD patients and non-PD control subjects. They reported a relationship between BR hypoechogenicity and the presence of depression, with an inverse correlation between the grade of echogenicity and the severity of depression (ρ = −0.646, *p* < 0.001). They also found a significant enlargement of the lateral ventricles compared to non-depressed PD patients. In 1999, Berg et al. [12] analyzed RMI and ultrasound midbrain changes in 31 PD patients, they found that BR echogenicity was significantly reduced in depressed PD patients, which was consistent with the findings previously reported. However, no correlation between midbrain intensity in RMI and BR echogenicity was demonstrated. Since then, many authors have studied brain parenchy mal ultrasound characteristics and related them to NMS, especially to the presence of de pression. BR hypoechogenicity, found in 35 to 85% depressed PD patients compared to 6 to 27% in controls, was associated with concomitant depression in PD patients in all revised studies [11,12,13,14,15,16,18,19,20,21,22], except for one which involved 126 early stage PD patients and compared BR and SN alterations between depressed (only 16 out of the 72 included subjects) and non-depressed patients based on the Hamilton Depression Rating Scale [17]. In addition, BR hypoechogenicity was also more frequent in non-PD patients with unipolar depression [14,18,20]. Interestingly, when compared to healthy controls, non-depressed PD patients showed no differences in BR echogenicity [11,14,15,18,19,20,21]. Most of the studies reported a correlation between BR hypoechogenicity and the severity of depressive symp toms independently of age, disease duration, and Hoehn and Yahr stage [11,15,16,18,19,21,22]. One study analyzed platelet serotonin levels as a biomarker of depression and correlated them with the TCS findings, without evidencing a significant relation [20]. Apathy, pessimistic thoughts, and anxiety were also related to BR hypoechogenicity [16,21].

In a study by Walter et al. [14] with 200 patients, 45 PD without depression, 45 PD with a depressive syndrome, and 110 non-PD patients, 55 of them with depression, SN hyperechogenicity was found in 40% non-PD patients with depression, 69% PD without depression and 87% depressed PD subjects, while it was only found in 3% of healthy controls. Non-Parkinsonian subjects with depression had a 3-fold higher frequency of SN hyperechogenicity compared to controls. Moreover, the combination of marked SN hyperechogenicity and reduced raphe echogenicity was significantly associated with a history of depressive disorder prior to onset of PD and with motor asymmetry in non-PD subjects with depression [14].

#### 3.3.2. Dementia

Four studies focused on the link between midbrain ultrasound changes and cognitive impairment or dementia in PD patients. They included a total of 375 PD subjects with and without dementia, 54 patients with other Parkinsonism, 14 patients with dementia with Lewy bodies (DLB), and 40 healthy controls [13,23,24,25]. Frontal horn dilatation and third ventricle dilatation were associated with dementia and the width of both ventricles corre lated with age but not with PD duration. No differences were identified between PD patients without dementia and controls [13,23,24,25]. Walter et al. [13] found that PD subjects with dementia had larger third ventricle width (8.7 ± 2.1 vs. 6.9 ± 2.5 mm; *p* = 0.002) and frontal horn width (17.3 ± 3.1 vs. 14.9 ± 3.1 mm; *p* = 0.003) compared to PD patients without dementia. Frontal horn was found to discriminate dementia in PD slightly better (AUC, 0.70; *p* = 0.006) than third ventricle (AUC, 0.69; *p* = 0.007), with a proposed cutoff value ≥ 15.4 mm for 82% sensitivity and 58% specificity [13].

In addition, based on the ROC curve, Dong et al. [25] suggested that a third ventricle width cut-off of 6.8 mm had a 69.6% sensitivity and a 61.5% specificity for discriminating between PD patients with and without dementia.

No differences in SN sizes were found in PD patients with dementia compared to those without dementia, both showing a larger SN than healthy controls [13,24,25]. The study of SN was useful to discriminate between DLB and PD, based on SN asymmetry and echogenic size [23].

Interestingly, in the group of atypical Parkinsonism, a significantly higher frequency of hypoechogenic BR was described in subjects with cognitive impairment compared to atypical Parkinsonisms without cognitive impairment [24].

#### 3.3.3. Autonomic Dysfunction

For the purpose of this review, the term autonomic dysfunction comprises all the symptoms derived from organs mainly dependent on the autonomic nervous system, such as constipation or urinary incontinence, even if the neurological mechanisms responsible for these symptoms in PD patients are not fully clarified and may present a central, peripheral, or combined pathophysiological mechanism.

Regarding urinary symptoms, Walter et al. [26] studied TCS characteristics (SN echogenic size and BR, thalami, lenticular nuclei and heads of caudate nuclei echogenicity, and widths of third ventricle and of frontal horns of lateral ventricles) in 116 PD patients divided into two groups, PD patients with overactive bladder symptoms (OAB) and PD patients without OAB symptoms, assessed by a clinical interview with the neurologist. Alternative etiologies of OAB were ruled out. BR hypoechogenicity was more pronounced in subjects with longer duration of urinary symptoms, with no other differences identified in the rest of the analyzed structures. Other authors evaluated the relation of midbrain transcranial structures and autonomic specific items in the Non-Motor Symptoms Questionnaire for patients with PD (PD-NMSQ), as well as the Scale for Outcomes in PD, autonomic symptoms (SCOPA-AUT) [32,33]. The PD-NMSQ consists of 30 items that address nine domains including gastrointestinal, cardiovascular, and urinary symptoms, sexual function, cognition (apathy, attention, memory), presence of hallucinations, depression or anxiety, sleep disorders, pain, and fatigue [34]. The SCOPA-AUT includes 25 items assessing autonomic symptoms: gastrointestinal, urinary, cardiovascular, thermoregulatory, pupillomotor, and sexual dysfunction [35]. None of the included studies reported any relevant relation between PD-NMSQ and SCOPA-AUT scores and the US findings [32,33].

In recent years, four studies have been published evaluating the vagus nerve diameter and cross-sectional area (CSA) in the cervical region by high resolution ultrasound, comparing between PD patients and healthy subjects [27,28,29,36]. In three of them, NMS were assessed with the Unified Parkinson’s Disease Rating Scale, part I (UPDRS I) [37], which has four questions concerning intellectual impairment, thought disorder, depression, and motivation/initiative [27], and with the PD-NMSQ [28,29]. Electrocardiographic heart rate variability was also analyzed as a marker of vagal cardiac innervation [29]. Walter et al. [29] found significant bilateral atrophy of the vagus nerve without differences in the spinal accessory or the phrenic nerves in PD patients compared to age-matched controls. Moreover, bilateral vagus nerve CSA correlated negatively with the PD-NMSQ total score (*r* = −0.51; *p* = 0.001) and with the sum score of autonomic items of the PD-NMSQ (*r* = −0.46; *p* = 0.003). Heart rate variability correlated only with the right vagus nerve CSA (*r* = 0.58; *p* = 0.001) [29]. Pelz et al. [28] obtained similar results regarding vagus CSA but did not demonstrate correlation with the PD-NMSQ. Contrary to this, Fedtke et al. [27] found no differences in the SCA in both groups.

#### 3.3.4. Restless Leg Syndrome

The TSC findings of 107 PD patients, 81 subjects with idiopathic restless leg syndrome (iRLS) and 75 age- and sex-matched healthy controls were analyzed in two studies [30,31]. SN echogenicity was significantly decreased in iRLS patients and increased in PD− RLS. Likewise, iRLS SN was significantly hypoechogenic compared to healthy controls [30,31]. No differences in SN were found between PD patients with and without RLS [30,31].

#### 3.3.5. Hallucinations and Psychosis

Walter et al. [13] found an association between the caudate nuclei hyperechogenicity and the presence of drug-induced psychosis in a group of 101 PD subjects. This finding was independent from PD duration. More recently, Li et al. [33] compared the TCS findings in a group of 111 PD patients and 61 non-PD controls, evaluating the presence of NMS with the PD-NMSQ, the Parkinson’s disease sleep scale, addressing sleep and nocturnal disability [38], the constipation severity instrument [39], and the Parkinson’s disease Fatigue Scale [40]. They reported, for the first time, that the SN echogenic area in PD patients with visual hallucinations (VH) was significantly higher than in those without VH. This finding was constant after adjusting by age, disease duration, and Minimental State Examination and UPDRS scores [33].

## 4. Discussion

The present systematic review aimed to provide a comprehensive analysis of the available literature reporting ultrasound findings in NMS in PD population. Given the expanding awareness of non-motor complaints in PD, we believe that such a review was necessary to better delineate the usefulness of ultrasound in the diagnosis and understanding of those features.

### 4.1. Brainstem Raphe

In the revised literature, BR hypoechogenicity has been related with depressive states in PD patients [11,12,13,14,15,16,18,19,20,21,22]. Reduced echogenicity of BR was more frequent in PD patients (25 to 30%) compared to controls (6 to 9%) [13,14], with a 3.5 higher risk of developing depression compared to non-depressed PD population [16]. This alteration had been previously reported for non-Parkinsonian patients with unipolar depression and depressive mood disorders, with a prevalence of 50–70% [41,42] and was confirmed in the studies comparing PD with depression and depressed patients without PD [14,18,20]. The same TCS pattern has been described in depressive patients suffering from other neurological conditions such as Wilson’s disease and Huntington disease [43,44], but not in multiple sclerosis [45,46].

There is no consensus about the association between BR hypoechogenicity and the severity of depressive symptoms, although some groups showed a negative significant correlation. In depressed non-PD patients, the presence of BR hypoechogenicity seems to predict a better response to serotonin reuptake inhibitors with 70% sensitivity, 88% specificity, and 88% positive predictive value [42].

The BR sonographic findings could be correlated to an increase in the signal intensity of the brainstem midline (raphe) on T2-weighted images on MRIs performed in depressed patients with and without PD [12], suggesting a structural disruption of the BR. These similarities in depressive patients with and without PD support the hypothesis of a structural alteration of the mesencephalon with a common pathophysiological basis [45].

Anatomically, the echogenic midline represents various nuclei and fiber tracts connecting serotonergic, dopaminergic, and noradrenergic brainstem nuclei with subcortical and cortical brain areas. The dorsal raphe nucleus is one of the BR structures and is considered the major origin for serotonin release in the brain [47]. A reduced echogenic signal of the BR could be due to alterations in the micro-architecture of this region, confirmed by a few histological reports, reflecting a serotonin deficiency which is involved in depression pathophysiology [45]. Depressed PD patients have low concentrations of serotonin, dopamine, and noradrenalin or their metabolites in cerebrospinal fluid [48]; this has been compared to platelet serotonin levels as a peripheral biomarker for depression, without success [21,49].

Serotonergic systems affection has also been proposed as a cause of overactive bladder in PD, activity in the serotonergic pathway generally enhances sympathetic innervation tone and detrusor hyperreflexia [50]. Epidemiological studies in humans have suggested an association between urinary incontinence and depression [51]. In agreement with these reports, Walter et al. [26] found a significant hypogenic BR in PD patients with OAB. Moreover, there was a greater number of OAB patients suffering from dysthymia or major depressive disorder. In line with these data, in a recent study by Roy et al., mean MRI diffusivity in the ventral brainstem, in areas close to the pontine micturition center and the pontine continence center, correlated significantly with the bladder symptom severity in PD patients [52].

### 4.2. Substantia Nigra

Not only raphe hypoechogenicity but also SN hyperechogenicity has been related to increased liability of depression, in both PD and non-PD populations [14]. Non-PD depressed patients present a 3-fold increased frequency of presenting SN hyperechogenicity [14]. This finding could be interpreted as a risk marker for PD development, supported by epidemiological studies that evidenced an increased risk of PD development in depressive patients [53,54,55]. Furthermore, the co-occurrence of SN hyperechogenicity and BR hypoechogenicity in PD patients was associated with history of depression prior to PD onset [14].

Another interesting finding of Li et al. [33] involving SN hyperechogenicity was its relationship with the presence of VH, evaluated with the PD-NMSQ. The exact pathogenesis of VH in PD patients is not clearly understood. Based on brain imaging studies, an abnormality and dysfunction in visual cortex and cholinergic structures such as the SN and pedunculopontine nucleus have been proposed [56], in line with TCS findings. Caudate nucleus hyperechogenicity was also found to be related to drug-induced psychosis. The relation between SN hyperechogenicity and other NMS such as constipation, fatigue, and the presence of restless legs syndrome (RLS) has also been evaluated, without finding any difference between PD patients suffering or not from the evaluated symptom. Interestingly, when comparing RLS in PD and iRLS, a significant reduction of SN was found in iRLS. This suggests that, despite both having a good response to levodopa, a different pathophysiological mechanism may be involved.

### 4.3. Lateral and Third Ventricle (Width)

Previous reports found that elderly non-PD individuals with enlarged SN performed worse in neuropsychological tests than individuals with normal SN echogenicity; however, that was not confirmed in the analyzed publications [57,58].

The main finding correlated with cognitive impairment was enlarged lateral and third ventricles, not present in PD patients without dementia and healthy controls [13,23,24,25]. Those changes are not specific of PD and have been reported in other neurodegenerative dementias. Previous studies have described a pattern of brain atrophy which is similar in Alzheimer’s disease (AD) and in PD. Nevertheless, the cognitive impairment profile is different in both diseases; while AD predominantly affects memory, PD is characterized by the involvement of executive functions. At a pathological level, dementia in PD is thought to be secondary to Lewy body deposits in the neocortical and limbic system. However, pathological changes normally associated with Alzheimer’s disease, such as abnormal deposition of β-amyloid and neurofibrillary tangles, have been additionally proposed to contribute to dementia in some PD patients [59,60].

### 4.4. Vagus Nerve Atrophy

Finally, the most recently explored neurological structure is the vagus nerve diameter and CSA, assessed at the cervical level with high resolution ultrasound. The interest of this structure is based on the hypothesis that the vagus nerve may represent one major route of disease progression in PD, with an active retrograde transport of α-synuclein originating in the enteric nervous system, ascending the vagus nerve, and eventually reaching the dorsal motor nucleus of the vagus in the lower brainstem [61,62,63].

Four studies evaluated the vagus nerve in PD and non-PD patients, with inconsistent results [27,28,29,36] regarding higher susceptibility of this long nerve to α-synucleinopathies. Only one proved a relation between vagus nerve atrophy and the presence of NMS and an increase in hearth rate variability [29]. A few reasons could explain the varying results, including differences in methodology and clinical heterogeneity of the PD group, although age and UPDRS-III scores, addressing motor examination, were similar.

In this review, we found some limitations, such as the possible variability of the ultrasound evaluation protocols between the study groups. Furthermore, TCS is dependent on the examiner’s skill and examinations are limited by variables such as the acoustic bone window and angulation of the scanning plane. Moreover, due to the characteristic physical features of PD patients, blindness to diagnosis might be difficult to achieve. Regarding the assessment of NMS, mainly based on scales and anamnesis due to biomarkers scarcity, it is possible that interindividual self-perception variability may have been a limitation.

## 5. Conclusions

The results of this systematic review support the use of transcranial ultrasound as a valuable complementary tool in the evaluation and diagnosis of the main NMS in PD. Future studies assessing US characteristics in non-PD patients with NMS and evaluating the risk of developing PD as well as the response to medical treatment are needed.

## Figures and Tables

**Figure 1 brainsci-11-00776-f001:**
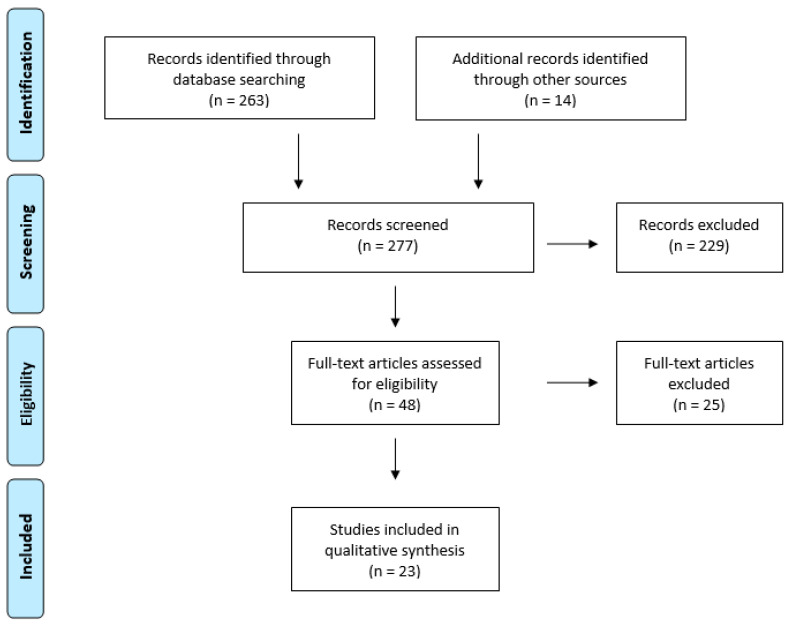
PRISMA Flow Diagram.

**Figure 2 brainsci-11-00776-f002:**
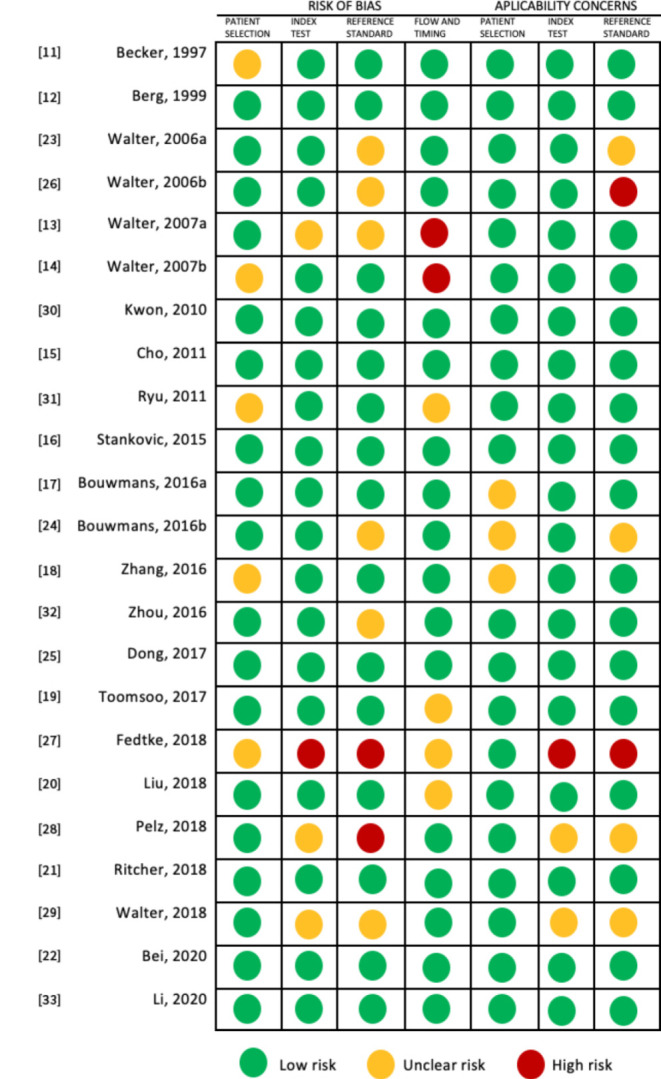
Assessment of risk of bias of studies. QUADAS-2 tool. QUADAS-2, Quality Assessment of Diagnostic Accuracy Studies-2.

**Table 1 brainsci-11-00776-t001:** Study characteristics.

STUDY	Population N, Age/Male	Symptom Evaluation	Ultrasound Evaluation	Main Findings	Other Findings
*Depression*					
Becker, 1997 [11]	30 PD+. 68,3/25 30 PD– 65/24	DSM-III HDRS CGI-S	TCS, 2.25 MHz. BR echogenicity * Ventricles Width	PD+, D+: ↓BR echogenicity, ↑lateral ventricles. Correlation: BR echogenicity and D severity.	No differences PD+, D− and healthy controls.
Berg, 1999 [12]	31 PD+ 65,5/16	DSM-IV HDRS BDI	TCS, 2.5 MHz. BR echogenicity *	PD+, D+: ↓BR echogenicity	MRI: PD+, D+: hyperintense mesencephal ic midline
Walter, 2007a [13]	101 PD+ 66,6/58	DSM IV	TCS, 2.5 MHz. BR echogenicity ^	PD+, D+: ↓ BR echogenicity	N.R.
Walter, 2007b [14]	45PD+, D+ 45PD+, D− 55 PD−, D+ 55PD−, D− 61/84	DSM IV DRS BDI	TCS, 2.5 MHz. SN echogenicity (N < 20 mm^2^) BR echogenicity ^.	PD+, D+: ↓BR echogenicity. PD+, D+ vs. D−: No difference in SN. ↑SN, ↓BR: Depression prior to PD diagnosis	PD−, D+: ↑SN
Cho, 2011 [15]	61 PD+ 68/38 41 PD−, D– 58/28	HDRS BDI	TCS, 2.5 MHz. BR echogenicity *	PD+, D+: ↓BR echogenicity, Correlation: ↓ BR echogenicity and ↑Hamilton Depression Scale.	PD + D+: higher motor severity
Stankovic, 2015 [16]	118 PD+ 61/72	HARS Apathy Scale MADRS	TCS, 2.5 MHz SN echogenicity (N < 19 mm^2^) BR echogenicity *	PD+, D+: ↓BR echogenicity (> sadness, pessimism, >anxiety)	↓BR echogenicity, ↑L-Dopa motor complications.
Bouwmans, 2016a [17]	72 PD+, 68/N.R. 54 other PK 72/N.R.	HDRS	TCS, 2–4 MHz. SN echogenicity (N < 20 mm^2^) BR echogenicity ^ 3rd. ventricle Width	No differences (Only 16 D+)	N.R.
Zhang, 2016 [18]	80 PD+ 40 PD− D+ 40 PD– D− 61/97	HDRS BDI	TCS, 2.5 MHz BR echogenicity ^†^	PD+, D+ and PD−, D+: ↓BR echogenicity. Correlation: ↓ BR echogenicity and ↑HDRS, BDI.	N.R.
Toomsoo, 2017 [19]	266 PD+ 168 PD– 69,7/228	BDI	TCS, 1.8–3.6 MHz SN echogenicity (N < 20 mm^2^) BR echogenicity *	PD + D+ and PD− D +: ↓BR echogenicity. Correlation: ↓ BR echogenicity and ↑BDI	Correlation: D and PD duration, motor and cognitive impairment.
Liu, 2018 [20]	30 D+ PD+ 30 D− PD+ 24 D+ PD−	HDRS	TCS, 2.5 MHz BR echogenicity ^ SN echogenicity	PD+, D+ and PD− D+: ↓BR echogenicity. No	Platelet serotonin.
	28 D– PD− 55/56	Platelet serotonin levels	(N < 20 mm^2^). 3rd. ventricle width	association SN and RN echogenicity.	Levels: no differences.
Ritcher, 2018 [21]	31 PD+ 16 ET+ 16 PD−, ET−	Lille apathy rating scale. BDI	TCS, 2.5 MHz. SN echogenicity (N < 20 mm^2^) BR echogenicity *	PD+: ↓BR echogenicity. Correlation: ↓ BR echogenicity and ↑Apathy, Beck Scores.	No difference: SN in ET+ and controls.
Bei, 2020 [22]	135 PD+ 63/83	HDRS HARS	TCS, 2.5 MHz SN echogenicity (N < 20 mm^2^) BR echogenicity ^	D+, Anxiety+: ↓BR echogenicity. Correlation: ↓ BR echogenicity and ↑Hamilton Scores, PDQ-39	No relation BR and motor symptoms.
*Dementia*					
Walter, 2006a [23]	104 PD+ 14 DLB+ 70/69	MMSE Addenbrooke cognitive examination	TCS, 2.5 MHz SN echogenicity (N < 20 mm^2^) Thalami, Caudate, BR echogenicity ^, 3rd. ventricle Width	PD + Dementia+: ↑ lateral frontal (17.3 mm), 3rd ventricle (8.6 mm) widths.	DLB+ vs. PD+ dementia +: Bilateral ↑ SN in DLB+. Similar ventricle widths.
Walter, 2007a [13]	101 PD+ 66,6/58	DSM IV MMSE	TCS, 2.5 MHz. ventricles width	PD + Dementia+: Lateral frontal horn ≥15.4 mm	↑Caudate echogenicity: ↑drug- induced psychosis.
Bouwmans, 2016b [24]	72 PD+ 68/70 54 other PK 72/80	SCOPA-COG: PD cognition Scale.	TCS, 2–4 MHz. SN echogenicity (N < 20 mm^2^) BR echogenicity ^, 3rd. ventricle width	Larger 3rd ventricle in PD+ and cognitive impairment. SN: Not related to cognition.	Atypical PK + cognitive symptoms: ↓BR echogenicity (not in PD)
Dong, 2017 [25]	98 PD+ 77/68 40 PD– 65/27	Dementia clinical diagnosis. MMSE MoCA PD-NMSQ	TCS, 2.5 MHz. SN echogenicity (N < 20 mm^2^) 3rd. ventricle width (Normal < 7/10 mm under/over 60 y.)	Larger 3rd ventricle in PD+ with dementia. Cutoff 6.8 mm (S: 69.6%, Sp: 61.5%). SN: Not related with cognition.	3rd. ventricle: No differences PD without dementia and controls.
*Autonomic dysfunction*					
Walter, 2006b [26]	116 PD+ 66,5/65	Overactive bladder symptoms (other causes ruled out)	TCS, 2.5 MHz. BR echogenicity * SN echogenicity, thalamus, 3rd.ventricle width	Overactive bladder: ↓BR echogenicity.	N.R.
Fedtke, 2018 [27]	32 PD+ 30 PD− 70/40	UPDRS I–IV	HRUS Vagus nerve (cervical CSA)	No differences PD+, PD−. No correlation with UPDRS I-IV.	Positive correlation: Right CSA and bradykinesia score.
Pelz, 2018 [28]	35 PD+ 35 PD– 67/34	PD-NMSQ, MoCA	15 MHz HRUS Vagus nerve (cervical CSA)	PD+: Smaller bilateral CSA. No	N.R.
*correlation with PD− NMSQ.*					
Walter, 2018 [29]	20 PD+ 73/13 61 PD− 45/23	PD-NMSQ, heart rate variability (R-R)	15 MHz HRUS Vagus, spinal, accessory, phrenic nerves (cervical CSA)	PD+: Smaller bilateral CSA. Negative Correlation: CSA and PD-NMSQ, autonomic items. Heart rate variability and right CSA in PD + and PD−. No differences in other nerves	Left CSA correlates with motor severity.
*Restless legs syndrome*					
Kwon, 2010 [30]	63 PD+ 65/30 40iRLS+ 53/21 40 controls 69/21	Sleep questionnaire Neurologist assessment	TCS, 2.5 MHz. SN echogenicity (N < 20 mm^2^)	SN: No differences in PD + with and without RLS.	iRLS+: ↓↓SN size than PD+ and controls.
Ryu, 2011 [31]	44PD+ 41iRLS+ 35 controls 60–71	RLS Diagnostic criteria	TCS, 2.5 MHz. SN echogenicity (N < 20 mm^2^)	SN: No differences in PD + with and without RLS.	iRLS +: ↓↓SN size than PD+ and controls.
*Hallucinations and psychosis*					
Zhou, 2016 [32]	201 PD+ 92 PD– 60/193	PD-NMSQ Odor test RBDSQ SCOPA-AUT MMSE HDRS	TCS, 2.5 MHz. SN echogenicity (N < 18 mm^2^)	No correlation SN and non-motor symptoms	Correlation: SN and UPDRS-II score
Li, 2020 [33]	111 PD+ 61 PD– 66;63/110	PD-NMSQ Sleep Scale, Constipation, Fatigue, MMSE HDRS, HARS	TCS, 1.82 MHz. SN echogenicity (N < 23.5 mm^2^)	PD with hallucinations: ↑ SN echo-size	No other differences

BDI: Beck’s Depression Inventory, BR: Brainstem Raphe, CGI-S: Clinical Global Impression-Severity scale, CSA: cross-sectional area, D+: depression, D-: no depression, DLB+: dementia with Lewy bodies, DSM: Diagnostic and Statistical Manual of Mental Disorders, ET+: essential tremor, ET-: no essential tremor, HARS: Hamilton Anxiety Rating Scale, HDR: Hamilton Depression Rating Scale, HRUS: high resolution ultrasound, iRLS: idiopathic restless legs syndrome, MADRS: Montgomery–Asberg Depression Rating Scale, MMSE: Minimental State Examination, MoCA: Montreal Cognitive Assessment, N: normal reference value, N.R.: not reported, PD+: Parkinson’s disease, PD−: No Parkinson’s disease, PD-NMSQ: Non-Motor Symptoms Questionnaire for patients with Parkinson’s disease, PK: Parkinsonism, RBDSQ: Rapid Eye Movement Sleep Behavior Disorder Screening Questionnaire, iRLS+: idiopathic Restless Leg Syndrome, SCOPA-AUT: Scale for outcomes in PD of autonomic symptoms, SN: Substantia nigra, TCS: transcranial sonography, UPDRS: the Movement Disorder Society-sponsored revision of the Unified Parkinson’s Disease Rating Scale. ↑ Increase or improved, ↓ Decreased or worsened. * Three grades semiquantitative scale using the hyperechogenic red nucleus as a reference point: 1 = raphe not visible/isoechogenic raphe compared with adjacent brainstem parenchyma, 2 = slightly echogenic raphe, 3 = normal raphe echogenicity (echogenicity of the raphe is identical to that of the red nucleus). ^ Two grades semiquantitative scale: 1 = not visible or interrupted echogenicity; 2 = continuous echogenicity compared to red nucleus. ^†^ Four grades semiquantitative scale: 1 = invisible raphe, echogenic raphe was not visible; 2 = interrupted raphe, echogenic raphe was interrupted compared with the red nucleus; 3 = decreased raphe, echogenic raphe was decreased compared with the red nucleus, but it was continuous; 4 = normal raphe, with the same echogenicity of red nucleus. Grades 1–3 are determined as abnormal.

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
