# Peer review of "Neurosonological Findings Related to Non-Motor Features of Parkinson’s Disease: A Systematic Review"

_brainsci, 2021, doi:10.3390/brainsci11060776_

Round 1

Reviewer 1 Report

The paper is well prepared but only the expert reader is able to follow the descriptions of NMS their evaluations and the neurosonological findings, perhaps the paper could include some transcranical sonography and HRUS images to explain what you are speaking about to include a larger readership

Also some explanation on the scales which are described in the paper. this as a paper addressed to a very specific audience and not understandable for a large reader ship, examples of sonography and explanation caould be added also in an appendix  

Author Response

We thank the reviewers for their comments and welcome the opportunity to clarify a number of points.

Response to reviewer 1:

The paper is well prepared but only the expert reader is able to follow the descriptions of NMS their evaluations and the neurosonological findings, perhaps the paper could include some transcranical sonography and HRUS images to explain what you are speaking about to include a larger readership

Also some explanation on the scales which are described in the paper. this as a paper addressed to a very specific audience and not understandable for a large reader ship, examples of sonography and explanation caould be added also in an appendix  

Following the reviewer’s suggestion, we have added explanations of the main scales used to evaluate non-motor symptoms as well as an appendix including ultrasound images that explain the main transcranial structures evaluated.

Reviewer 2 Report

This is a systematic review of observational studies with the aim to present the most common neurosonological findings in patients with Parkinson’s disease and non-motor features.

Although this is an interesting review, there are several issues that need to be further addressed by the authors.

  1. Is there a registration number available for PROSPERO?
  2. Authors may consider presenting their inclusion criteria regarding study design as well. What kind of studies were considered eligible? Did they include both cohort studies and case series as well? “Observational” for the characterization of a study design is not enough.
  3. Table 1 is quite overwhelming and difficult to follow with all the abbreviations. Also, the authors should consider presenting the population of interest consistently in every study. For example, in Waltersb, both PD+D+ and PD+D- are presented, whereas this is not the case for Becker, Berg etc.
  4. In paragraph 3.1, “dysautonomia” should precede “dementia” since it was more commonly evaluated.
  5. Please improve the quality of Figure 2. Specifically, please correct the underlined studies.
  6. In paragraph 3.3.1, where authors state that “BR hypoechogenicity, found in 35 to 85% depressed PD patients compared to 6 to 27% in controls, was associated with concomitant depression in PD patients in all revised studies (11-22) except for one”, they should actually remove the citation #17, since no association was noted in this study.
  7. In paragraph 3.3.1, where authors state that “One study analyzed platelet serotonin levels as biomarker of depression and correlated them with the TCS findings, without evidencing a significant relation”, an appropriate citation should be provided.
  8. The statement “Ultrasound is a safe and low-cost technique that, although operator-dependent, can easily complement the clinical evaluation supporting the clinician's diagnosis” cannot be presented as a conclusion of the study, since neither safety nor cost-effectiveness were evaluated in this review. This is a comment that should have been better included in the Discussion section. Please revise the conclusion, presenting only the main findings of this review together with a future perspective.
  9. Language needs significant editing. Several typos exist throughout the manuscript.

Author Response

We thank the reviewers for their comments and welcome the opportunity to clarify a number of points.

Response to reviewer 2:

This is a systematic review of observational studies with the aim to present the most common neurosonological findings in patients with Parkinson’s disease and non-motor features.

Although this is an interesting review, there are several issues that need to be further addressed by the authors.

  1. Is there a registration number available for PROSPERO?

The protocol of this systematic review follows the guidelines according to the preferred reporting items for systematic reviews and meta-analysis protocol (PRISMA-P) and was registered in PROSPERO international database of prospectively registered systematic reviews (CRD 42021250195).

  1. Authors may consider presenting their inclusion criteria regarding study design as well. What kind of studies were considered eligible? Did they include both cohort studies and case series as well? “Observational” for the characterization of a study design is not enough.

Cross-sectional observational, case-control and cohort studies including patients with Parkinson disease, ultrasound assessment of neurological structures and evaluation of NMS, analyzing differences between echogenicity and/or size of the evaluated structures between PD suffering or nor from a specific NMS. Case reports were excluded.

  1. Table 1 is quite overwhelming and difficult to follow with all the abbreviations. Also, the authors should consider presenting the population of interest consistently in every study. For example, in Waltersb, both PD+D+ and PD+D- are presented, whereas this is not the case for Becker, Berg etc.
  2. In paragraph 3.1, “dysautonomia” should precede “dementia” since it was more commonly evaluated.

Following the reviewer’s suggestion, we have added reduced the number of abbreviations and presented the population of interest consistently in every study.  Likewise we have corrected the total number of included studied assessing dementia, a total of four.

  1. Please improve the quality of Figure 2. Specifically, please correct the underlined studies.

The quality if Figure 2 has been corrected as recommended by the reviewer.

  1. In paragraph 3.3.1, where authors state that “BR hypoechogenicity, found in 35 to 85% depressed PD patients compared to 6 to 27% in controls, was associated with concomitant depression in PD patients in all revised studies (11-22) except for one”, they should actually remove the citation #17, since no association was noted in this study.
  2. In paragraph 3.3.1, where authors state that “One study analyzed platelet serotonin levels as biomarker of depression and correlated them with the TCS findings, without evidencing a significant relation”, an appropriate citation should be provided.

The citation mistakes have been properly corrected.

  1. The statement “Ultrasound is a safe and low-cost technique that, although operator-dependent, can easily complement the clinical evaluation supporting the clinician's diagnosis” cannot be presented as a conclusion of the study, since neither safety nor cost-effectiveness were evaluated in this review. This is a comment that should have been better included in the Discussion section. Please revise the conclusion, presenting only the main findings of this review together with a future perspective.

In agreement with the reviewer’s assessment, the conclusion of the manuscript has been revised and future perspective have been presented. We have included the following paragraph to the main text:

The results of this systematic review support the use of transcranial ultrasound as a valuable complementary tool in the evaluation and diagnosis of the main NMS in PD. Future studies exploring US characteristics in non-PD patients with NMS and assessing the risk of developing PD as well as the response to medical treatment are needed.

  1. Language needs significant editing. Several typos exist throughout the manuscript.

Following this recommendation, the present manuscript underwent English revision.

We thank you for your interest in our research and your suggestions to improve the quality of the manuscript. We hope that these clarifications have resolve any doubts concerning our article and the revised manuscript will better suit Brain Sciences.

Round 2

Reviewer 2 Report

Authors have appropriately addressed all issues raised during the first round of review. No further comments exist on my behalf.